# Bioactive Metabolites of Marine Origin Have Unusual Effects on Model Membrane Systems

**DOI:** 10.3390/md18020125

**Published:** 2020-02-19

**Authors:** Martin Jakubec, Christian Totland, Frode Rise, Elahe Jafari Chamgordani, Britt Paulsen, Louis Maes, An Matheeussen, Lise-Lotte Gundersen, Øyvind Halskau

**Affiliations:** 1Department of Biological Sciences, University of Bergen, Thormøhlensgate 55, NO-5006 Bergen, Norway; martin.jakubec@uib.no; 2Department of Environmental Chemistry, Norwegian Geotechnical Institute, Sognsveien 72, 0855 Oslo, Norway; christian.totland@ngi.no; 3Department of Chemistry, University of Oslo, P.O. Box 1033, Blindern, NO-0315 Oslo, Norway; frode.rise@kjemi.uio.no (F.R.); elahe.jafari2009@gmail.com (E.J.C.); britt.paulsen@kjemi.uio.no (B.P.); 4University of Antwerp, Laboratory of Microbiology, Parasitology and Hygiene, Faculty of Pharmaceutical, Biomedical and Veterinary Sciences, Universiteitsplein 1, B-2610 Antwerp, Belgium; louis.maes@uantwerpen.be (L.M.); an.matheeussen@uantwerpen.be (A.M.)

**Keywords:** agelasine, malonganenone, phase behavior, lipids, membrane affecting drug candidates, ^31^P NMR

## Abstract

Marine sponges and soft corals have yielded novel compounds with antineoplastic and antimicrobial activities. Their mechanisms of action are poorly understood, and in most cases, little relevant experimental evidence is available on this topic. In the present study, we investigated whether agelasine D (compound **1**) and three agelasine analogs (compound **2**–**4**) as well as malonganenone J (compound **5**), affect the physical properties of a simple lipid model system, consisting of dioleoylphospahtidylcholine and dioleoylphosphatidylethanolamine. The data indicated that all the tested compounds increased stored curvature elastic stress, and therefore, tend to deform the bilayer which occurs without a reduction in the packing stress of the hexagonal phase. Furthermore, lower concentrations (1%) appear to have a more pronounced effect than higher ones (5–10%). For compounds 4 and 5, this effect is also reflected in phospholipid headgroup mobility assessed using ^31^P chemical shift anisotropy (CSA) values of the lamellar phases. Among the compounds tested, compound 4 stands out with respect to its effects on the membrane model systems, which matches its efficacy against a broad spectrum of pathogens. Future work that aims to increase the pharmacological usefulness of these compounds could benefit from taking into account the compound effects on the fluid lamellar phase at low concentrations.

## 1. Introduction

The scientific and commercial interest for finding new compounds with antibiotic and antineoplastic properties has increased in the last two decades as antibiotic and multi-drug resistance in cancer remains a pressing societal challenge [1,2,3,4]. Approaches to meet this challenge include producing analogs of existing compounds and designing new compounds that interfere with the cellular processes that are required for proliferation [1,5]. Another approach is bioprospecting for new compounds with interesting biomedical properties, and from these develop analogs that bypass multidrug resistance.

Marine bioprospecting has provided many compounds of interest, particularly from demosponges [6,7,8,9,10]. Certain marine sponges and gorgonians (sea fans; soft corals) produce bioactive metabolites that can be regarded as hybrids between a diterpenoid and a purine derivative. These compounds are adenine or (hypo) xanthine derivatives, often carrying the terpenoid substituent at the N-7 position of the purine ring. Representative examples are the agelasines [11] and malonganenones [12], such as agelasine D (**1**) and malonganenone J (**5**) (Figure 1).

Agelasines are associated with antineoplastic, antimicrobial activities (including against tuberculosis, malaria, and leishmaniasis), biofilm inhibition, as well as having antifouling properties [11,13,14]. We have previously identified cytotoxic properties for several agelasine analogs. These analogs simplified the side chains and also involved modifications in the adenine ring [15,16,17,18,19,20]. Data for analogs (**2**–**4**) are shown in Table 1. We have also previously reported the first total synthesis of a malonganenone J (**5**) [21] and evaluated its biological activity (Table 1). However, the mechanisms of these activities are not well understood.

The agelasines have been found to affect a number of adenosine–triphosphate dependent transmembrane ion pumps, including P2X7, Na^+^, K^+^-ATPase, and sarcoplasmic Ca^2+^-ATPase [22,23,24,25,26]. For malonganenones, the following targets have been reported: tyrosine kinase c-Met (malonganenone D [27]), phosphodiesterase PDE4D (malonganenone L [12]), and plasmodial heat shock protein PfHsp70-1 (malonganenone A [28]). Except for the dioxygenase, all the above targets have binding pockets accepting adenosine moieties. It is tempting to speculate that interference with these proteins is central to their mode of action (MOA). While MOA evidence exists for individual cases [26,28,29], a comprehensive picture has yet to emerge.

Here, we investigated an alternative or complementary MOA for agelasine and malonganenone compounds: the perturbation of the lipid membrane. The overall structural features of compounds **1**–**5** are the purine heterocycle with an appended diterpenoid (aliphatic chain or chain-fused ring system). The compounds resemble cell membrane lipids, where a polar head group is attached to an elongated aliphatic moiety. This resemblance may have a bearing upon their in vivo activity mechanism. For example, it is conceivable that such compounds have a strong physical as well as biological interaction with cellular membranes. This would imply that such compounds are also cytotoxic because they intercalate in, interfere with, or even disrupt the structure of cellular membranes. Biological activities that depend on the compound’s ability to interact with the bilayer could also include access to targets buried in the plasma membrane. In this scenario, the compound would intercalate within the membrane and then interfere with ligand binding sites of the aforementioned P-type ATPases or lipid flippases that are buried in the core of the bilayer. Such membrane-related mechanisms are reported to exist for other lipid-like or otherwise membrane-active drugs, whereby the lipid membrane is increasingly considered as a potential drug target itself [30]. One notable example, the alkylphospholipids, has found niche uses in cancer treatment [5]. The sea anemone hemolytic proteins cytolysin (I and II) and equinatoxin II act through pore formation directly on the lipid membrane [31,32,33]. It has also been documented that transferring oleic acid to the cell membrane using protein fatty-acid complexes, liprotides, cause cancer cell death through disruptive effects on the membranes [34,35]. These phenomena, along with recent evidence indicating that even small changes to the composition of lipid systems may have a pronounced effect on their phase behavior [29,36,37,38], prompted us to investigate malonganone J, agelasine D, and agelasine analogs using hydrated dioleoylphosphatidylcholine (DOPC) and dioleoylphosphatidylethanolamine (DOPE) model lipid systems doped with these compounds (**1**–**5)**.

DOPC is typically a bilayer lipid [39] that can be used to investigate changes in stored curvature elastic stress and fluidity. DOPE re-assembles into the inverse hexagonal phase above 281 K [40] and can be used to investigate changes in packing stress of the inverse hexagonal phase and in stored curvature elastic stress. The effect of compounds **1**–**5** on phase behavior was explored through temperature scans (273–338 K) of doped DOPC and DOPE systems using wide line ^31^P NMR. The experimental data were deconvoluted to yield information about the relative amounts of phases present, and CSA values, which give information about headgroup mobility [41,42], were extracted.

## 2. Results

All the compounds (**1**–**5**) were synthesized locally based on previously published techniques [15,16,21]. The minimum inhibitory concentration (MIC) of compounds **1**–**4** has already been determined for several pathogens [15,16,17]. To supplement these, we performed MIC assays on compound **5**, on bacteria and protozoa, such as *Staphylococcus aureus* and *Trypanosoma cruzi*. The IC_50_ value for the latter was fairly low at 0.9 µg/mL. The other bioactivity values are summarized in Table 1, alongside with previously published data. Reviewing these data did not reveal any obvious pattern within compound **1**–**5** that is relatable to their structural variation. Low MICs or IC_50_ values did not appear to favor either the branched-chain or fused carbocycles configuration of the aliphatic moiety, nor any of the three different heterocyclic systems. The latter observation is at odds with a strictly adenosine-based activity, and alternative or complementary MOAs should be sought. Concerning the lipid-like configuration of compounds **1**–**5**, we investigated how these molecules affected simple models of the cellular bilayer.

DOPC and DOPE are representatives of two bulk lipids found across practically all eukaryotes [43,44,45,46], with PE also being common in prokaryotes [37,47]. The phase behavior of DOPE and DOPC has been investigated in considerable detail [39,40,42,48,49], making deviations from the typical behavior of these systems generally straightforward to identify [29,38,50,51]. Therefore, they are useful for characterizing the behavior of lipids in response to minor additives [29,36,38]. Doping with 1%, 5% and 10% of synthetically pure samples of compounds **1**–**5** was done to characterize modulations in the lipid–lipid interactions in the DOPC and DOPE assays. Equilibrated hydrated samples were scanned over a temperature range of 273–338 K, and wide-line ^31^P NMR spectra were collected for each temperature. The spectra were then deconvoluted by simulation into lamellar, isotropic, and inverse hexagonal components (Figure 2A and Appendix A). This made it possible to determine and to some extent, quantify the occurrence of new phases. Other parameters, such as the chemical shift anisotropy (CSA) values of the lamellar phases, were estimated based on the deconvoluted spectra. The CSA parameter of ^31^P nuclei in hydrated lipids is related to the mobility of the headgroups [41,52]. From the temperature scans, we observed how additives perturbed phase transitions relative to the pure situations (e.g., the lamellar-inverse hexagonal phase transition at 281 K in DOPE [40]). Taken together, we could then determine whether a given additive changed stored curvature elastic stress (SCES), the lamellar phase headgroup mobility, or the packing stress of the inverse hexagonal phase.

There was a low-intensity signal consistent with an inverse hexagonal phase in DOPE samples containing compounds **1, 3**, and **4**(~6 ppm, marked * in Figure 3A)) at 1% abundance at 273 K, with only agelasine D (**1**) having the same effect at higher concentrations (10%, Figure 3B). This is notable because the transition from the fluid lamellar phase to the inverse hexagonal phase in hydrated DOPE is at 281 K [40]. For contrast, full DOPE inversion at 310 K and 10% addition is presented in Figure 3D. Moreover, the isotropic phase was visible in mixtures of all five compounds, even at 273 K and at 1% in the systems based on the bilayer-forming lipid DOPC (Figure 3C). These data suggest that compounds **1**–**5** promote the formation of non-bilayer phases, but in most cases, have little impact on packing stress. Despite clear evidence for an increase in SCES, none of the compounds was able to induce reassembly of the entire lamellar DOPC system; all systems retained a clear fluid lamellar phase in addition to the isotropic phase (Figure 3C). Increasing concentration did not increase the occurrence of an isotropic phase in DOPC; instead, it stabilized the lamellar phase and increased the transition temperature of DOPC (Appendix A).

The tendency for compounds **1**–**5** to induce the formation of an isotropic phase over the lamellar phase in DOPE is **4** > **3** >> **2** ~ **1** > **5** (Figure 3D). The higher SCES of compound **4** compared to agelasine D (**1**) matched the considerable difference between **4** and **1** for IC_50_/MIC values in eukaryotic systems, suggesting that these may be vulnerable to this curvature-forming species at low concentrations. This also links with the lowering effect the 1% addition of **4** had on the CSA parameter of DOPC lamellar phases. There were less diverse biological activity data available for compound **5**, which had a similar but less clear-cut CSA effect. However, for protozoa, both these compounds display a low IC_50_ (<1 µg/mL).

## 3. Discussion

Our results suggest that the compounds had notable low-abundance effects on lipid bilayers, in particular deformation of the fluid lamellar (bilayer) phase (compounds **1, 3, 4**) and increased headgroup mobility (compounds **4** and **5**) at low concentration. All compounds induced isotropic phases in DOPC, but only compound **4** and **3** did so strongly in DOPE. Only compound **4** was notable in all these regards (Table 1), and it was effective, to some extent, against the pathogens tested. It may be that the unusual membrane effects exerted by this compound’s display could explain its broad efficacy, as both lowering the membrane CSA and inducing isotropic or inverse phase components would be detrimental to cell membrane integrity. Indeed, there are many studies that link cytotoxicity or drug action to such mechanisms, including the alkylphosplipids [5], protein-fatty acid complexes delivering cytotoxic oleic acid to the membrane [54,55], and numerous membrane-active proteins and peptides [31,33,52,56]. Environmental toxicity of pollutants has also been ascribed to membrane effects in some cases [41]. Often, both direct action on the membrane as well as protein targets are reported, and the relatively new field of membrane–lipid drug therapy recognizes both [30].

The low-concentration effects, including the appearance of non-lamellar phases and lowered CSA values, suggest that the compounds elicit complex changes in the lipid environment. It seems that when their concentrations increase, the interactions between the additives themselves help to stabilize the lamellar phase. As the concentration of compounds decreases, so do available partners for drug-to-drug interaction, and the compound starts to have an inverse effect: by disturbing the lipid bilayer and reassembling it into an isotropic phase. The possibility of self-inhibitory effects is important in the investigation of drug dosing [57]. The exact nature of the lipid organization giving rise to the isotropic ^31^P NMR resonance, including whether the compounds partition preferentially into them or not, was not determined in this study. In principle, micelles, small vesicles, and bicubic phases may all give rise to isotropic ^31^P NMR resonances. It may be possible to differentiate between these situations using ^31^P relaxation measurements on model systems prepared explicitly for this purpose [58].

Regardless of the exact nature of the observed phase changes, our results suggest that purine–terpene hybrid natural products can damage lipid membrane structures at therapeutically-relevant concentrations. Compounds **1** and **2**, each with a cationic methyladeninium head group, appeared to be less effective at inducing non-lamellar phases with their respective geranylgeranyl moieties. However, the two analogs which were strongest at effecting SCES (**4** and **3**) and the least effective analog (**5**), all comprise a geranylgeranyl moiety. This suggests that the combination of the head group (purine building block) and the lipophilic terpenoid side chains both play a role in drug activity, rather than just either one of them. It may help the compounds to retain broad bioactivity profiles, and may reflect whether a given compound relies more on membrane perturbation or protein targets in a given case. More mechanistic studies are undoubtedly needed, and it may be that further drug development guided by the compounds’ effect on lipid membranes may improve the outcomes of such work.

## 4. Material and Methods

*Reagents & Chemicals—*Solvents and fine chemicals were purchased from Sigma–Aldrich (Gillingham, Dorset, UK) and used without purification. Synthetic lipids were purchased from Avanti Polar lipids Inc. (Alabaster, Alabama, AL, US) and used without purification. Compounds **1**–**5** were synthesized according to literature procedures [15,16,21]. Antimicrobial data for compound **5** were obtained as described before [59].

*^31^P NMR sample preparation—*Lipid/additive mixtures were made in solution (dichloromethane: methanol, 1:1, 10–15 mg/mL) and dried to a lipid film in vacuo. Dried lipid films were dispersed in aqueous buffer (NaCl 100 mM, tris 50 mM, CaCl_2_ 2.5 mM, MgCl_2_ 2.5 mM, pH 7.4; 10:1 *v/w*) with sonication and agitation. Deuterium oxide (10% *v/v* against buffer) was added, and the mixture agitated and then freeze–thawed 8–10 times (193–313 K). Samples were then stored at 193 K, transported at ~295 K and stored at 253 K before running.

*Wide line**^31^**P NMR spectroscopy—*A Bruker Avance III HD 400 MHz spectrometer equipped with a Bruker BBO S1 (smart) probe was used for proton-decoupled wide line ^31^P NMR experiments. Experiments were performed at 161.98 MHz with an inverse-gated pulse sequence with proton decoupling during acquisition, the spectral width of 200.44 ppm, the acquisition time of 0.299 s, pre-scan delay (DE) time of 20 µs, receiver gain of 203 with 19,428 data points and 2048 scans per spectrum. The duration of each scan, acquisition time and relaxation delay was 2.332 s. Spectra were processed using TopSpin 3.2 and 3.5 with line broadening of 50 Hz, phase correction and automatic baseline correction (abs).

*^31^P NMR temperature scans—*Samples were equilibrated at the desired temperature for 15 min before the acquisition. Temperature scans consisted of acquisitions at the following temperatures: 298, 273, 293, 298, 310, 318, 273, 310 K. The traces from the latter two temperature points were overlaid on the earlier traces at the same temperature and the fit used to assess the reliability of the equilibration time used (15 min). Acquisitions at 338 K were carried out separately with equilibration and acquisition at 310 K immediately beforehand. Stack plots of spectra were prepared manually from individual traces processed as above.

*Wide line ^31^P NMR data deconvolution—*^31^P NMR spectra deconvolution was carried out using the Bruker Topspin software package and the *sola* interface (Version 3.5). Figures were plotted in MATLAB 2017 b. Up to three functions were fitted to the experimental data using the Gauss/Lorentz alternative in the interface. Typically 90–95% of the data could be accounted for in this way. The fraction relative to the total signal of each phase for a given fitting, as well as accompanying CSA values extracted from the fits, were noted.

## Figures and Tables

**Figure 1 marinedrugs-18-00125-f001:**
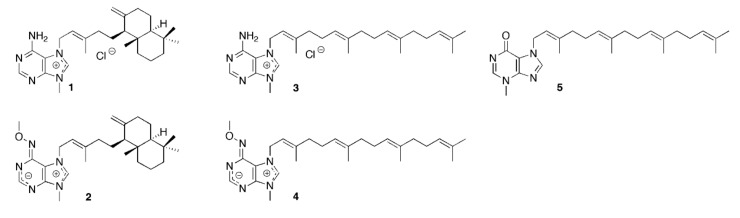
Molecular structures of (+) Agelasine D (**1**), agelasine analogues (**2**–**4**), and malonganenone J (**5**).

**Figure 2 marinedrugs-18-00125-f002:**
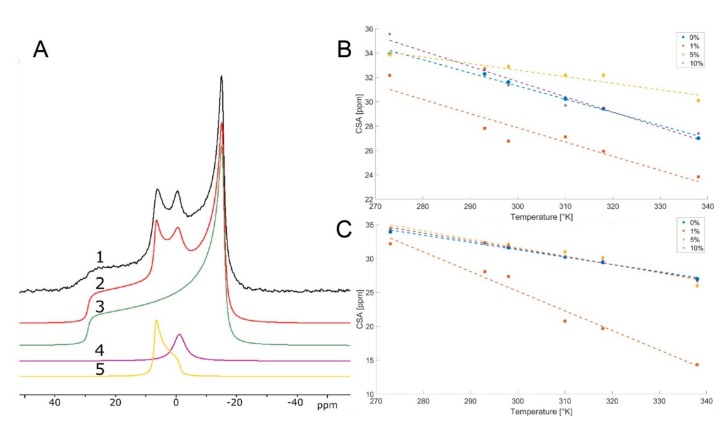
Deconvolution and chemical shift anisotropy (CSA) parameter extraction of selected hydrated lipid samples. (**A**) Representative deconvolution of ^31^P wide line NMR data of hydrated dioleoylphosphatidylethanolamine (DOPE) at 273 K, with a 10% addition of compound **1**. Individual traces represent 1—Acquired wide line spectra, 2—Sum of fitted shapes, 3—Fit of lamellar phase, 4—Fit of isotropic phase, 5—Fit of inverse hexagonal phase. (**B**) CSA of dioleoylphosphatidylcholine (DOPC) lamellar phase after addition of 0%, 1%, 5%, or 10% of compound **4**. Dashed lines show linear fits of CSA changes with increasing temperature. (**C**) CSA of DOPC lamellar phase after addition of 0%, 1%, 5%, or 10% of compound **5**. Dashed lines show linear fits of CSA changes with increasing temperature. Upon extracting the CSA parameters associated with the lamellar phases, it is possible to plot them as a function of temperature and additive percentage [53]. Generally, and as expected, the CSA parameters fall as temperatures increase (Figure 2B,C). For pure DOPC at 273 K, the CSA of the ^31^P nuclei in their headgroups are 34 for DOPC (Figure 2). This value suggests a significant orientation and restriction. Only for compound **4** and **5** in DOPC, we see a significant perturbation of the CSA values. Intriguingly, the 1% addition has the strongest effect (Figure 2B,C).

**Figure 3 marinedrugs-18-00125-f003:**
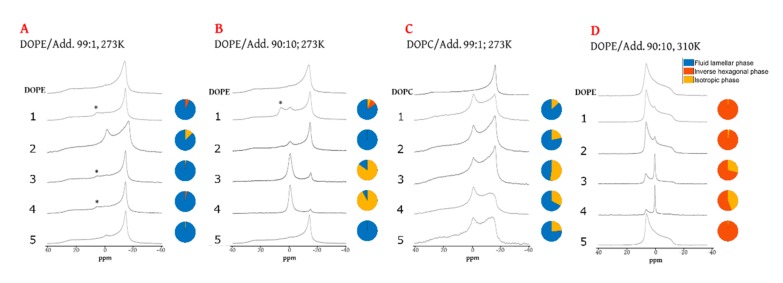
Line ^31^P NMR traces of lipid systems doped with compounds **1**–**5** and controls at 273 K. Panel (**A**), DOPE with 1% additive; (**B**), DOPE doped with 10% additive; (**C**), DOPC doped with 1% additive; (**D**), DOPE doped with 10% additive at 310 K. Traces taken from temperature scans (see Appendix A) run with samples at strict equilibrium. Deconvoluted spectra are in Appendix A. * Signal position consistent with an inverse hexagonal phase in DOPE.

**Table 1 marinedrugs-18-00125-t001:** Inhibitory activities for compounds 1-5 against microorganisms and cancer cell lines

Compound	Bacteria MIC (µg/mL)								Fungi MIC (µg/mL)	Cancer Cells IC_50_ (µg/mL)				Protozoa IC_50_ (µg/mL)			
	*Staphylococcus aureus*	*Streptococcus pyogenes*	*Enterococcus faecalis*	*Escherichia coli*	*Pseudomonas aeruginosa*	*Bacteroides fragilis*	*Bacteroides thetaiotaomicron*	*Mycobacterium tuberculosis*	*Candida krusei*	U-937GTB	RPMI8226/s	CEM7S	ACHN	*Plasmodium falciparum*	*Leishmania infantum*	*Trypanosoma cruzi*	*Trypanosoma bruceii*
1	1 ^c^	2 ^c^	8 ^c^	8 ^c^	16 ^c^	16 ^c^	8–16 ^c^	>6.25 ^c^ 92% at 6.25 µg/mL	–	2.6 ^c^	1.8 ^c^	2.3 ^c^	11.8 ^c^	0.3 ^a^	1.5 ^a^	4.5 ^a^	0.9 ^a^
2	2 ^c^	2 ^c^	8 ^c^	8 ^c^	32 ^c^	4–8 ^c^	4–8 ^c^	>6.25 ^c^ 96% at 6.25 µg/mL	–	0.5 ^c^	0.1 ^c^	1.1 ^c^	3.7 ^c^	0.3 ^a^	0.6 ^a^	0.5 ^a^	0.3 ^a^
3	32 ^b^	–	–	>32 *^d^*	–	–	–	>6.25 ^b^ 38% at 6.25 µg/mL	>16 ^b^	1.5 ^b^	1.3 ^b^	1.5 ^b^	9.6 ^b^	–	–	–	–
4	4 ^b^	4 ^b^	8 ^b^	16 ^b^	>32 ^b^	8 ^b^	8 ^b^	3.13 ^b^	2.0 ^b^	0.7 ^b^	0.5 ^b^	0.9 ^b^	3.6 ^b^	0.1 ^a^	0.1 ^a^	0.1 ^a^	0.2 ^a^
5	32 *^d^*	–	–	>32*^d^*	–	–	–	–	–	–	–	–	–	–	4.6 *^d^*	0.9 *^d^*	4.5 *^d^*

Inhibitory activity of compounds **1**–**5**. Full species names: *Staphylococcus Aureus, Streptococcus Pyogenes, Enterococcus Faecalis, Escherichia Coli, Pseudomonas Aeruginosa, Bacteroides Fragilis, Bacteroides Thetaiotaomicron, Mycobacterium Tuberculosis, Candida Krusei, Plasmodium Falciparum, Leishmania Infantum. Trypanosoma Cruzi, Trypanosoma Brucei*. ^a^ Data reproduced from Vik et al., **2009** [15]; ^b^ Data reproduced from Vik et al., **2007** [16]; ^c^ Data reproduced from Vik et al., **2006** [17]; ^d^ new data.

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
