# Peer review of "Bioactive Metabolites of Marine Origin Have Unusual Effects on Model Membrane Systems"

_marinedrugs, 2020, doi:10.3390/md18020125_

Round 1

Reviewer 1 Report

Review of Jakubec et al

The manuscript entitled “Small Additions of Bioactive Marine Sponge Metabolites Have Unusual Effects on Lipid Headgroup Mobility and Phase Organization in Membrane Model Systems” by Jakubec et al provided new insights into the potential of certain marine metabolites on lipid membrane structures

Number of inaccuracies, unclear sentences, sentences out of context that disturb description and discussion, you will find them marked in the pdf file Title is too long..

References are tool old What was the rationale to select those five compounds? This study is suitable as short communication

The biomedical action resulting from effects of Agelasines on the lipid model system is relatively unclear, and should be clarified in the abstract.

Author Response

Review of Jakubec et al

The manuscript entitled “Small Additions of Bioactive Marine Sponge Metabolites Have Unusual Effects on Lipid Headgroup Mobility and Phase Organization in Membrane Model Systems” by Jakubec et al provided new insights into the potential of certain marine metabolites on lipid membrane structures

Author Response 00: We thank the reviewer for his input. Our point-by-point responses are listed below in italics.

Number of inaccuracies, unclear sentences, sentences out of context that disturb description and discussion, you will find them marked in the pdf file Title is too long..

Author Response 01: Practically all suggestions by the reviewer have been implemented. The manuscript has then received a thorough round of copy-editing to ensure good flow, precision and clarity. The title has been shortened from 21 words to 12 words. We have also modified Figure 1 and converted units in Table 1, in accordance with reviewer request.

References are tool old What was the rationale to select those five compounds? This study is suitable as short communication

Author Response 02:

References: The paucity of knowledge on these metabolites required use of also older references in some cases. Also, lipid biophysics is a mature field where it is sometimes correct to cite seminal papers, even though they are old. This notwithstanding, we have included two recent (2019) papers in the references. In the updated manuscript, more than half of the references are from 2010 or more recent. Many are from the last couple of years. Only a couple are from before 2000.

The rationale for selection these compounds were 1) that they have interesting bioactivities, 2) that their MOA is as yet uncertain, and 3) that they share a vague similarity to lipids. They also were selected because the authors have developed synthetic pathways for them in earlier work.

Transfer from "Article" to "Short Communication": It appears that Marine Drugs do not have this type of article. However, we have requested that the manuscript will be published as a short communication, and have used Short Communication articles published by other MDPI journals as a template when preparing the revision. This did not have a large impact on the organization of the manuscript, though.

The biomedical action resulting from effects of Agelasines on the lipid model system is relatively unclear, and should be clarified in the abstract.

Author Response 03: We have copy-edited the abstract for greater clarity.

Reviewer 2 Report

This manuscript presents the effect of marine sponge metabolites on phospholipid mobility. The work seems carefully done and the introduction, experiment design, analyses, results, and conclusions are clearly presented.

The manuscript needs corrections as follows.

(1) Line 45: ‘(compound 1)’ should be corrected to ‘(1)’.

(2) Line 45: ‘(compound 5)’ should be corrected to ‘(5)’.

(3) Material and Methods: The authors should describe about the reference for the 31P NMR chemical shifts.

Author Response

This manuscript presents the effect of marine sponge metabolites on phospholipid mobility. The work seems carefully done and the introduction, experiment design, analyses, results, and conclusions are clearly presented.

Author Response 00: We thank the reviewer for his kind words, and for his or her time and effort. Our responses are listed below in italics.

The manuscript needs corrections as follows.

(1) Line 45: ‘(compound 1)’ should be corrected to ‘(1)’.

(2) Line 45: ‘(compound 5)’ should be corrected to ‘(5)’.

Author Response 01: These two changes have been implemented.

(3) Material and Methods: The authors should describe about the reference for the 31P NMR chemical shifts.

Author Response 02: There are only two different lipids used in this study, DOPC and DOPE. Here they are present as hydrate multilamellar structures, and as such there is no single ppm-value associated with them. Rather, wide line NMR displays a continuous range of shifts that are dependent on the distribution of molecular orientations present in the sample, relative to the B0 field of the spectrometer. It is possible to provide references for ppm values for these lipids in organic solvents, but that would be irrelevant or even misleading, so we do not explicitly do this. Citation 37, 45 and 47 do have shift information for PC and PE species in a commonly used solvent, however. These references also highlights the combined use of wide line/broad line NMR and liquid state 31P NMR. Readers who need more information on the theoretical background and physical significance of the extracted parameters from wide-line NMR are also referred to citation 41 and 42 in the updated manuscript.

Round 2

Reviewer 1 Report

manuscript has been improved